# Changes in the development of opioid tolerance on re-exposure among people who use heroin: A qualitative study

Joanna May Kesten[1,2,3]*, Ed Holder[4], Rachel Ayres[4], Pete Ellis[4,5], Steve Taylor[6], Matthew Hickman[1,3], Graeme Henderson[7]

1 NIHR Health Protection Research Unit in Behavioural Science and Evaluation, University of Bristol, Oakfield House, Oakfield Grove, Bristol, United Kingdom, 2 NIHR Applied Research Collaboration (ARC) West at University Hospitals Bristol and Weston NHS Foundation Trust, Bristol, United Kingdom, 3 Population Health Sciences, Bristol Medical School, University of Bristol, Bristol, United Kingdom, 4 Bristol Drugs Project, Bristol, United Kingdom, 5 Developing Health and Independence, Brunswick Court, Bristol, United Kingdom, 6 Alcohol & Drugs Treatment & Recovery, Office for Health Improvement and Disparities, Department of Health and Social Care, London, United Kingdom, 7 School of Physiology, Pharmacology & Neuroscience, University of Bristol, University Walk, Bristol, United Kingdom

* Jo.Kesten@bristol.ac.uk

**Data Availability Statement:** The data are available at the University of Bristol data repository, data. bris, at https://doi.org/10.5523/bris.23rgdwwmdtix52gqa51ith8gxy. Data access is

## Abstract

### Background and aims

This qualitative study aimed to explore how the development of tolerance to both the psychoactive and respiratory depressant effects of heroin on re-exposure are experienced by people who use heroin.

### Methods

Semi-structured one-to-one interviews were conducted with 20 adults who currently or previously used heroin (for at least 6 months), with any type of administration (injected, smoked) and experience of abstinence (at least 2 weeks) and relapse. Topic guides explored the participants understanding of tolerance, their experience of developing tolerance to heroin and of tolerance following relapse. Interviews were audio-recorded and transcribed. Thematic analysis was used to generate salient themes.

### Results

The analysis produced three broad themes: lay understanding of tolerance; tolerating tolerance; and rapid tolerance development following relapse. Tolerance was defined as the body adapting to regular drug use, so that the drug no longer produced the same level of effect. Tolerance was experienced as interacting and co-developing with physical dependence and the symptoms of withdrawal. Indeed, several participants did not differentiate between tolerance and dependence. Most participants did not notice tolerance to respiratory depression. Tolerance levels fluctuated—increasing over periods of regular use and reducing when abstinent. Using more drug was the most common response to increasing tolerance to the desired effects. On re-use following abstinence, tolerance was experienced as

restricted to bona fide researchers for ethically approved research and subject to approval by the University's Data Access Committee. The University Data Access Committee can be contacted on data-bris@bristol.ac.uk. Access to the data is restricted to researchers because there is a degree of sensitivity involved and the University's Ethics committee agreed to data sharing with access restrictions only. Furthermore participants consented to their data being shared with researchers. Participants were also expected to be less inclined to answer the questions fully if the data was shared openly.

**Funding:** JK and MH acknowledge support from the National Institute for Health and Care Research Health Protection Research Unit in Behavioural Science and Evaluation (NIHR HPRU in BSE) at University of Bristol and JK is partly funded by NIHR Applied Research Collaboration West (NIHR ARC West). ST is funded by the Office for Health Improvement and Disparities, Department of Health and Social Care (formerly Public Health England). GH is funded by the Medical Research Council MR/S010890/1. This study was funded by the Medical Research Council and the NIHR HPRU in BSE at University of Bristol, in partnership with UK Health Security Agency (UKHSA). The views expressed are those of the author and not necessarily those of the MRC, the NIHR, the Department of Health and Social Care, or UKHSA. There was no additional external funding received for this study.

**Competing interests:** The authors have declared that no competing interests exist.

developing more quickly in the most recent relapse compared to the first. Tolerance was also perceived to return more quickly with each abstinence-relapse cycle.

## Conclusions

Qualitative accounts of tolerance report that tolerance returns more quickly with each relapse episode. By elucidating the mechanism(s) involved and potentially discovering how they could be switched on prior to relapse occurring we might be able to develop a beneficial harm reduction treatment for people in abstinence that would protect against overdose occurring on subsequent relapse.

## Introduction

Intravenous use of illicit heroin is associated with an increased risk of premature death due to overdose and blood borne infections such as Human Immunodeficiency Virus (HIV) and hepatitis C virus (HCV) [1]. Drug-related deaths involving opioids are a public health emergency in North America and an emergent public health crisis in the United Kingdom (UK). In the UK, the number of heroin overdose deaths more than doubled from 825 to 2,020 between 2012–2019; and in the USA where fentanyls have become a major cause of opioid overdose deaths the number has grown exponentially with over 93,000 deaths in 2019 [2].

Heroin injection produces a range of effects that include euphoria, sedation, respiratory depression, analgesia, constipation and miosis [3]. Repeated administration over time of heroin or other opioids leads to the development of 'tolerance' to some but not all of their effects. Tolerance being defined as needing larger amounts of drug to feel the same effect or reduced effect to the same dose [4]. Whilst tolerance is undoubtedly a multi-faceted phenomenon [5,6] at a rudimentary level we can divide tolerance mechanisms in two: pharmacological and psychological (Fig 1A) [7]. Pharmacological tolerance, defined as a lowered response to the drug on repeated administration with the basal level remaining constant between drug doses [7]. This form of tolerance results primarily from the opioid receptors in the brain being 'switched off' by desensitisation, driven by phosphorylation of the receptors by specific kinases [8,9]. On the other hand, psychological tolerance may result from a number of different adaptive changes such as behavioural or conditioned learning induced by fear of going into withdrawal [6] as well as a progressive lowering of mood in the periods of withdrawal (development of a negative emotional state) between doses [7,10–12]. In this case, the response to the drug may not be diminished, but the peak effect attained is lower due to the lowering of the psychological baseline from which the drug response starts (Fig 1B).

It is likely that tolerance to the desired psychoactive effects of heroin–euphoria (rush), well-being and sedation (gouching)—is a combination of both pharmacological and psychological tolerance. White and Irvine [13] suggested in their theoretical analysis that tolerance to opioid-induced 'euphoria' developed to a greater extent than tolerance to respiratory depression although there is a lack of experimental evidence to substantiate this theory. If this difference is real then a potential explanation would be that tolerance to respiratory depression would involve only pharmacological tolerance as there is unlikely to be a depression of mood component to such tolerance.

Many people who use heroin cycle through periods of drug use, opioid substitution therapy (OST)—an evidence-based treatment [14,15] detoxification, abstinence and relapse, with a high proportion recommencing heroin use following a period of abstinence [16]. It is assumed

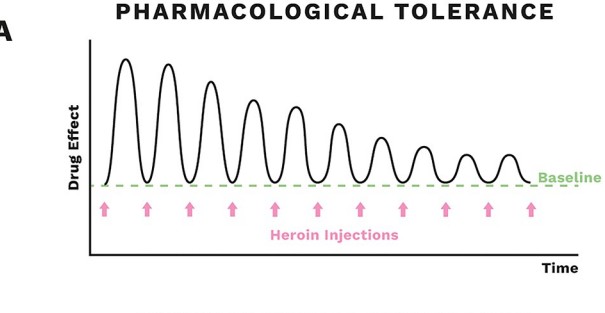

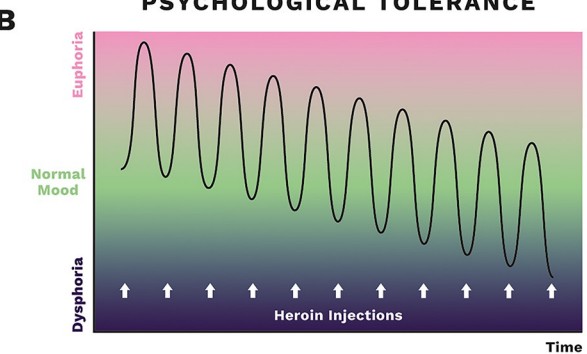

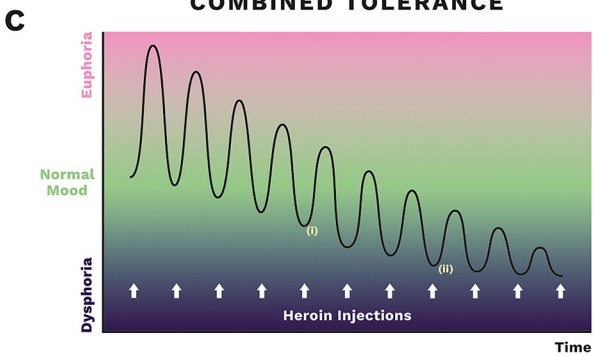

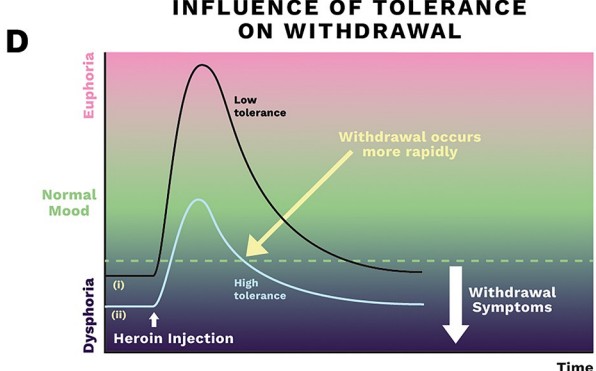

**Fig 1. Tolerance development.** A schematic showing the amplitude of response evoked by a repeated dose of heroin. (A) In the development of pharmacological tolerance, opioid receptors become progressively less responsive and this results in a reduced response amplitude with no change in baseline. (B) In psychological tolerance there is no change in the amplitude of the response evoked by the drug injection but between injections mood is depressed when the effect of the drug wears off. As the baseline mood falls progressively with drug use then the peak height of the drug effect is also lowered. Adapted from reference 11. (C) If both pharmacological and psychological tolerance are produced then not only does the baseline mood fall between drug doses but the amplitude of the response evoked by the drug injection will also decrease. (D) Comparison of the responses to the same dose of heroin under circumstances

of low and high tolerance as indicated by (i) and (ii) in parts C & D. The dashed line indicates the level below which withdrawal symptoms would appear. As indicated by the yellow arrow withdrawal is experienced more rapidly when tolerance is higher. Acknowledgement: Artwork by Roseanna Jackson of Slowe Club.

that loss of tolerance during abstinence is responsible for increased risk of mortality on relapsing back to heroin use [17–20].

In a preliminary conversation about tolerance development with a small group of experienced people who use heroin at the Bristol Drugs Project (BDP) people who use heroin asked why tolerance seemed to develop much faster on relapse following periods of abstinence compared to when they had first started using heroin. Subsequent literature searching revealed scant qualitative studies relating to experiences of tolerance [21–23] and a dearth of evidence on the rate and extent to which tolerance develops to the effects of heroin following a period of abstinence compared to its development when drug use was first initiated. The relative influence of psychological and neurochemical factors on tolerance development on re-exposure are also unclear. For example, psychologically, people may be more confident about drug use and take higher doses at greater frequency following relapse than compared to when they first initiated drug use, they may be more cautious, expecting their tolerance to have diminished with abstinence or they may deliberately take breaks to reduce their tolerance and feel a greater effect upon re-use. Alternatively, neurochemical adaptive changes may take place in the brain during the development of and subsequent decline in tolerance during drug use and detoxification that influence the rate and extent of tolerance development on relapse. While our future goal is to undertake reverse translational research (using animal experiments) to define these neurochemical changes in the brain, it is pertinent to first understand the lived experience of tolerance development over time and following relapse [24] and then build upon that to investigate potential biological mechanisms.

This qualitative study thus aimed to explore how the development of tolerance to both the desired psychoactive effects and respiratory depressant effects of heroin on re-exposure are experienced by people who use heroin. Specifically, how does tolerance compare when people first start using heroin regularly to after the most recent relapse and how do people manage changes in tolerance? To support our interpretation of the data, it was necessary to invite participants to define what tolerance means to them and how they experience the development of tolerance to both the desired psychoactive effects and respiratory depressant effects of heroin.

## Methods

We conducted semi-structured one-to-one interviews with people who use heroin.

### Sample and recruitment

The study was conducted in Bristol which has a large population of opioid users (13.48 per thousand of the population compared to English national prevalence of 7.33 [25]) in collaboration with BDP. BDP provides a range of services including a Needle and Syringe Programme (NSP), naloxone provision (overdose reversing medication), Mobile Harm Reduction, OST and physical healthcare [26].

Adult BDP service users were sampled for experience of using heroin for at least 6 months, current or within the past year use of heroin or OST (including methadone or buprenorphine/Subutex®), any type of administration (injected, smoked etc.), experience of abstinence from all opioids for at least 2 weeks–in an attempt to ensure they have experience a decline in tolerance—and experience of relapse. Although there is no agreed definition of time for tolerance to decline, UK clinical guidance for OST prescribing recommends that if a patient has missed

five days of their medication they need to be re-assessed and their dose may be reduced due to the possibility that their tolerance has reduced [27].

To gain access to potential research participants, BDP staff and volunteers, including staff at different sites within BDP delivering NSP and OST (the latter are known as shared care workers working in general practices across the city), distributed information sheets to potential participants and discussed the project with them. Interested service users were asked to either contact the lead researcher (JK) directly or speak to someone at BDP. The BDP recruiter then contacted JK on behalf of anyone willing to be interviewed. JK explained the study by phone or in person, answered any questions and confirmed eligibility prior to obtaining written informed consent. Recruitment procedures were adjusted throughout the study to obtain a diverse sample. During data collection, the recruiters were asked to purposefully sample individuals whose experience we had not captured, for example seeking younger individuals to add to the perspectives of older participants. The advantage of this approach to recruitment is that BDP staff have a rapport and trusting relationship with their service users and are likely to know service users who fit the recruitment criteria.

We aimed to interview 20–30 participants, varying in a range of demographic characteristics (e.g. age, gender, housing status). This sample size was based on previous experience of conducting qualitative research with BDP service users [28]. The decision to end data collection was informed by 'information power' which considers the focused nature of the aim, the sample specificity (characteristics of the participants relating to the phenomenon under study), quality and depth of the data and the analysis approach [29]. Data collection and analysis proceeded concurrently to assess the adequacy of the sample size and diversity of the participants.

## Study design

The study adopted an interpretative approach seeking to understand how people interpret and perceive their experiences and the social world and explore the actions people take in response to these experiences [30].

All aspects of the study were developed in collaboration with BDP co-authors and a group of BDP service users.

To address the study aims open-ended questions (S1 Appendix) were used to explore lay understanding and experiences of tolerance to heroin–changes in tolerance over time and following repeated relapse events compared to initial tolerance development. Participants were asked to focus on their experience of tolerance to heroin rather than dependence because this was the focus of the study. To explore respiratory depressant effects, participants were asked "have you experienced changes in the effect that drugs have on your breathing as a result of tolerance developing?" The guide was used flexibly and developed over subsequent interviews to allow the pursuit of emergent topics. Participants were asked to supply demographic information.

Interviews were audio recorded and transcribed verbatim. Participants received £10 cash as a thank you.

Ethical approval was obtained from the Faculty of Medicine and Dentistry Committee for Research Ethics, University of Bristol (Ref: 74821).

## Data analysis

Thematic analysis [31] was chosen to generate themes for its flexibility, transparent process and applicability to an interpretative approach [32]. Transcripts were read and re-read to gain familiarity with the data and preliminary impressions noted. Inductive codes were then

assigned systematically to the first few interviews grouped deductively under the main categories in the topic guide (e.g. 'Definition of tolerance', 'Experience of tolerance', 'Factors influencing tolerance' etc.). These codes were discussed among the research team, iteratively refined and combined to produce an agreed coding framework, which was further refined as subsequent interviews were coded. Frequent de-briefing between JK and GH ensured different interpretations of the data were considered. Final overarching themes and subthemes were agreed and descriptive accounts produced. Negative cases (highlighted in the findings) with deviating views or experiences were explored and compared to the rest of the dataset using the constant comparison method [33]. The use of semi-quantification is intended to highlight patterns in the data [34].

## Results

20 interviews lasting 18 minutes to an hour (40 minutes on average) were conducted between November 2018 –May 2019 (Table 1). Participants ranged from 19 to 54 years and had started using heroin regularly when they were 11–30 years, 4 were women, 6 had no fixed abode, all had injected heroin, 11 were currently using heroin (2 irregularly) and 17 were taking OST at the time of the interview. Some participants had experienced an overdose. The most common number of episodes of abstinence from heroin and OST was 3 to 4 with a range of 1 to 9. The longest time abstinent ranged from 2 weeks to 17 years.

Three themes were generated from the analysis: lay understanding of tolerance; tolerating tolerance; and rapid tolerance development following relapse.

### *"The amount of heroin that you consume before feeling it (*...*) and the amount that you feel it"*: Lay understanding of tolerance

Several participants defined tolerance as the body becoming 'acclimatised' so that the drug no longer gave the effect initially felt which meant greater quantities were needed to achieve the same 'high'. The drug's 'effect' also related to how long the 'high' lasts before needing to use again.

> *It's like getting into a bath of hot water. It's really hot at the start but within a minute or two you've acclimatised and you're no longer feeling the heat, you're just normalised in it. There's tolerance.* Interview 12

> *Tolerance is just how long it holds you for until your next hit.* Interview 9

Many participants depicted tolerance as interacting and co-developing with physical dependence and the symptoms of withdrawal between hits. Recognition of increasing dependence and tolerance over time were linked: *after about two weeks I would say I realised I had a problem with it (*...*) That's when I realised I've got a habit [dependence] to it and that's when I needed it and then I realised that after a while just one bag wouldn't do and I'd have to do two or three bags to get [high]–so my tolerance level's gone up* (Interview 14).

Similarly, several participants did not differentiate between tolerance and dependence, viewing them as synonymous and intimately related: *...It would take three or four days before my tolerance levels built up to a level where I was dependent* (Interview 20). Tolerance and dependence were perceived to have similar consequences resulting in a need for more drug to feel its effect and recover from withdrawal and similar trajectories: *creeps up on you* (Interview 15).

Worsening of withdrawal symptoms, needing more drug to alleviate them and, less time before they appear were seen as signs of increasing or high tolerance levels: *The higher your tolerance, the more you feel the effects of the withdrawal* (Interview 3). One participant described

**Table 1. Participant characteristics.**

| Characteristic | |
|---|---|
| **Age, years (mean, range)** | 39.55 (19–54) |
| **Gender (n)** | |
| Male | 16 |
| Female | 4 |
| **Housing status (n)** | |
| NFA (including sofa surfing and street homeless) | 6 |
| Housed (including temporary housing and full tenancy) | 14 |
| **Drugs currently used (n)** | |
| Heroin | 2 |
| Heroin and crack | 9 |
| Other drugs (cannabis, diazepam, ketamine, cocaine, Xanax, Spice) with or without heroin | 6 |
| **Age using heroin regularly (Mean age in years (range))** | 20.05 (11–30) |
| Never used regularly (n) | 1 |
| **Administration method (n)** | |
| Injected (n = 3 only injected) | 20 |
| Smoked (n = 15 smoked first then injected) | 17 |
| **Currently on OST (n)** | |
| Yes | 17 |
| Methadone | 9 |
| Buprenorphine / Subutex | 8 |
| No | 3 |
| **Number of abstinence episodes** | |
| **Times** | |
| 1–2 | 6 |
| 3–4 | 7 |
| 5–6 | 4 |
| 7–8 | 1 |
| 9 | 1 |
| Unclear / unspecified | 1 |
| **Length of abstinence** | |
| 2–6 weeks | 4 |
| 2–3 months | 2 |
| 9 months | 1 |
| 1–1.5 years | 2 |
| 2–3 years | 6 |
| 4–6 years | 3 |
| 14–17 years | 1 |
| Unclear reporting or unspecified | 1 |
| **Non-fatal overdose (n)** | |
| Yes | 8 |
| No | 7 |
| Unspecified | 5 |
| **Total** | **20** |

tolerance as a balance between the point at which withdrawal symptoms are removed while not feeling a 'high' from the drug. While another described not wanting to '*push*' their tolerance because this would result in greater dependence.

You get more dependent, you need it more because the effects on your body if you haven't got it, you feel a lot of pain, irritation, frustration.

Interviewer: So you're needing more?

*Participant: Yeah, [more] substance to cover it and bring you back to a normal level. But then once you get to that normal level then you want to go higher again.* Interview 7

Tolerance, dependence and the effects of drugs generally have a psychological aspect (i.e. believing you were becoming tolerant and expectations of the drugs effect):

*Your body recognises it and when you start using it your body's recognising it and accepting it a lot easier, mentally you're accepting it as well. Tolerance then builds up because you're expecting it psychologically as well, quite a lot.* Interview 7

*Psychological withdrawals always kick in earlier. If you're distracted, you're not going to notice them.* Interview 20

### *"As time went by my tolerance got higher and [I] started using higher and bigger amounts"*: Tolerating tolerance

**Inevitability and frustration.**   Participants typically reported low tolerance initially: *when I first started heroin my tolerance level was very low.* (Interview 14). When using regularly, many described tolerance to the desired effects as inevitably increasing over time (a few days to two years) depending on frequency of use. Only a few described tolerance stabilising, enabling them to use the same amount of drug repeatedly. After long periods of using heroin, the 'high' diminished completely; instead, using was necessary to feel 'normal' and remove withdrawal symptoms in what was described as a 'vicious cycle' (Interview 13): *At first it's nice and then it's just the more you do it the less it does and then you're just doing it because you've got to do it to feel normal and feel well because you're just going to be unwell if you don't do it* (Interview 5).

Becoming tolerant was described by a few as 'frustrating'—due to the need to take and spend more on drugs—and a barrier to a pleasurable high. One participant explained this frustration when he said: *tolerance to me is frustrating because of how quickly it seems to build up and just the fact that it's there (. . .), because I would have a much better quality of life if my tolerance was far, far less. (. . .) How would I define what tolerance is? (-) A barrier to getting to a satisfying level of intoxication* (Interview 8).

Several participants felt that poorer drug quality also contributed to the lessening effect of heroin rather than tolerance. Indeed, one participant felt that people who use drugs do not consider tolerance as an explanation for needing to use more: *Tolerance doesn't really enter into much of what junkies think, not really. All they think is bag, bag, bag, bag. Once they've got one, another one. They don't think about why they're needing more or anything like that. Most will just turn round and say 'well that gear was crap', not realising that tolerance has gone up.* Interview 15

**Unaware of breathing suppression.**   Most of the participants were unable to describe the development of tolerance to respiratory depression, largely because they were unaware of their breathing becoming depressed when they took heroin. Experiences of changes in breathing suppression following heroin use in relation to tolerance development, were ambiguous. A couple of participants described their breathing becoming shallower after taking heroin. Whereas others were not conscious of or did not notice changes to their breathing rate and a small number said they hadn't experienced these changes.

*I never, ever noticed is the breathing part really, no. You feel yourself relaxed but you don't consciously think I'm breathing slower really.* Interview 15

Overdose risk in relation to tolerance and the effect of the drug were considered. For example, one person explained how he balanced the quantity of heroin taken to achieve a high with the risk of overdose and another described intentionally taking enough heroin to get as close to overdosing as possible.

*Every day I probably would have taken a bit more just to try and get a higher high without getting too close to the overdose limit.* Interview 20

One participant talked about overdosing less frequently over time and one felt that increased tolerance levels explained why they hadn't overdosed.

**Responses to increasing tolerance.** Cumulatively using more drug to the point that it *"gets out of hand"* (Interview 5) was the most common response to increasing tolerance. Tolerance and the amount of drug used appeared to operate as a positive feedback loop: using larger quantities contributes to higher levels of tolerance, and greater tolerance leads to more drug being used. This pattern of increasing drug use in pursuit of the same effects as when they initially used was described as "insanity":

*Every time you're just chasing that initial feeling so you're just putting more and more and more on, then your tolerance builds and your habit gets worse and then it just escalates and it's just the same thing over and over again. It's just insanity really.* Interview 5

The time between hits of heroin also reduced and one person described using earlier in the day. There were a small number of accounts about attempting to maintain low tolerance and dependence levels for example by using 'periodically'. This meant less drug would need to be bought and used. One interviewee stated: *I don't push it too much more I suppose, I kind of try and keep it (. . .) I don't wanna push it any more 'cause I've got a big enough habit as it is. I don't wanna push it anymore and have a bigger habit. You know, it's hard enough to fund and control the habit that I have.* (Interview 29).

Generally, tolerance was perceived to increase more quickly when injecting compared to smoking, attributed to injected heroin entering the blood supply more rapidly. One participant described changing from smoking to injecting to feel a greater effect in response to tolerance developing:

*I went from smoking to IV [intravenous] because I wasn't getting the same effect and obviously, I felt like I built up a bit of a tolerance to it so I wanted something that would get me to the state I wanted to be in.* Interview 19

Some participants described tolerance and dependence prompting an intention to stop using heroin, recognising the body's limits and that using heroin was no longer enjoyable. The escalation in tolerance and dependence is captured in this account:

*You know I have to stop while it's [tolerance] low. You know I can't get–I can't get to the point where I'm scratching the itch 'cause if I was doing that from day one for three days, I'm gonna have to ween myself off. I mean I've just had a binge and I've just spent two grand in a month or just thereabout. So it goes quick, you know and I realised it was going up and I was like 'What the fucking hell are you doing?* Interview 2

**Influences on increasing dose and frequency of use.** The social and financial environment informs tolerance. A small number of participants commented that using heroin outside while experiencing homelessness, is more likely to perpetuate a use-score cycle as it is not easy to relax and the increased drug use contributes to greater tolerance levels. Two participants described other people's experience of the drug as influencing their expectations of its effect, when the same experience is not achieved this may encourage greater use.

*I used to always use at home. I used to use and then not have to go anywhere, just sit down and enjoy it and feel it, but when you're out in this sort of environment [on the street], you can't do that when you're out and about. (. . .) I think that's why a lot of people's habits are so bad as well when homeless and all that, because they don't get time to enjoy it so they just go straight on to get more which then could build your tolerance even higher.* Interview 10

**First use following a period of abstinence.** Reduced tolerance after a period of abstinence was common. When relapsing this resulted in perceptions of less drug being needed, greater drug effects and higher risk of overdose. The latter may be partly explained by advice from harm reduction workers and hearing of peers overdosing after abstinence.

*When I used again for the first time after my abstinence, it was very close to the first time, but it wasn't the same.* Interview 3

*You hear about friends coming out of prison, going straight out injecting and dying.* Interview 14

Reflecting on experiences of relapsing following a period of abstinence, participant accounts fell broadly into those which suggest people "err on the side of caution" and those which suggest people do not take precautions. Approximately half the participants described using less heroin (informed by assessing or asking about the quality of the drug) to test how their bodies respond and avoid overdose. However, the amount of drug used returned to pre-abstinence levels quickly. A less commonly reported practice was to inject a little at a time.

*I would have deliberately reduced it [amount of drug] by probably more than half to err on the side of caution, especially after a period of abstinence.* Interview 20

*I suppose I noticed it when I did stop for that little while and then just went to use what I was using before and realised I had to stop half way through doing my hit because I was like 'Woah that's–it kind of knocked me. I suppose I'm quite sensible as well, like I don't do it really fast so I can gauge how much it's gonna affect me.* Interview 19

Other precautions when relapsing included ensuring equipment is clean, not using every day, and smoking to lower the risks and then returning to injecting, while others immediately returned to injecting.

In contrast, other participants described using the same amount of drug following relapse. A few explained that this was due to poorer drug quality, i.e. less effect–this was also understood to explain why the same amount of drug had not caused an overdose. Two expressed a suicidal ideation or self-harm element to this: *I just jumped straight back in exactly as I was doing it before, feet first, don't care because my head's gone. (. . .) The thought that I might die. I don't care.* (Interview 6). For one interviewee who overdosed following relapse, their previously high tolerance was anticipated to mean they would "cope" with the same amount they used previously when relapsing, despite advice from harm reduction workers:

*I even got told an hour before I went and did it to be careful and not to do all at once, because I've been abstinent this amount of time, by the workers here [drug service], which is the mad thing like. And I still–because my past experience of my tolerance being high and being able to cope with it, I thought that it would be the same. (. . .) So, yeah, my tolerance obviously–psychologically I thought I'm the same. This gear's rubbish, don't do nothing, 'cause of the past times. No, it was too strong for me, and I nearly died.* Interview 3

### *"The more times I have a time of abstinence and then go back to it, the quicker the tolerance level shoots right up like a thermostat on a hot day"*: Rapid tolerance development following relapse

When asked to compare tolerance development over time, most participants experienced tolerance developing to the desired effect more quickly following a period of abstinence in the most recent relapse compared to the first. Similarly, a smaller number also perceived tolerance to return more quickly with each abstinence-relapse cycle.

*I've had periods of abstinence so obviously it's [tolerance] gone down but what I have found is my tolerance goes up very quickly now.* Interview 2

*I think that the first time that I developed tolerance took the longest and that every subsequent time took progressively shorter amounts of time, but that amount of time might occasionally be extended if the period of abstinence is greater.* Interview 20

Unlike most, one participant felt that tolerance did not return any quicker after the most recent relapse compared to the first, but attributed this to reduced drug quality.

There were mixed views on whether the length of abstinence influences the speed with which tolerance redevelops.

Several participants felt that tolerance returning quickly was due to the body remembering the drug effect, described by two participants as 'muscle memory' (muscles previously trained, strengthen more rapidly than untrained muscles [35])—and adapting to it quickly.

*Each time that I gets myself clean and I spend a period of time, like a couple of months (. . .) clean time, there seems to be a physiological change goes on within me, in my body. It's like each time my body says, "ah I know what's happening here, I know how to deal with this", and then your tolerance goes up that little bit more and it goes up that little bit quicker. It's like each time your body's learnt how to get to that point a bit faster.* Interview 12

*I've had periods of abstinence so obviously it's gone down but what I have found is my tolerance goes up very quickly now. It's almost like it's got–I've got a memory–my body's got a memory of it.* Interview 2

Two participants spoke about the influence of age on tolerance levels, with tolerance decreasing and the risk of overdose increasing with age following periods of abstinence.

*My bodily tolerance, my resistance to it has got less and less so the actual amounts I need to go over after a bit of clean time is pretty miniscule as I've got older.* Interview 6

It is unclear what underlies the re-development of tolerance. A theoretical explanation for why tolerance develops more rapidly on relapse, is that having experienced the development of tolerance previously, people may be expecting tolerance to develop when they relapse and are thus primed to move on to taking larger doses or begin with larger doses than they would have

done when they first started using heroin. This is illustrated by one participant relaying their thought processes around using after abstinence:

> *My tolerance obviously–psychologically I thought I'm the same. This gear's rubbish, don't do nothing, 'cause of the past times. No, it was too strong for me, and I nearly died like.* Interview 3.

Similarly, one participant spoke of the difficulty of knowing whether tolerance had developed and using in response to perceived tolerance and to avoid withdrawal: *As soon as I started using again after a period of abstinence, there's always the fear that tolerance has developed before it probably does. I think that very early on in my using I didn't know if I was developing tolerance and so quite often I would use simply because I thought I might have become tolerant and therefore it's better to use to avoid any illness that may or may not come about.* (Interview 20).

This would have a self-fulfilling effect–tolerance will in any event develop and if higher doses are taken then this will enhance even more tolerance development. Those who said that when they relapsed they used lower doses to avoid overdose still felt that tolerance developed quickly. Although participants did not express deliberately taking breaks from using heroin to reduce their tolerance and feel a greater effect upon re-use, accounts suggest attempts to maintain low tolerance levels.

As described earlier, the return of tolerance following relapse was intertwined with the re-development of dependence and withdrawal symptoms. For example, tolerance was believed by a small number of participants to have resulted in less sickness following relapse, thus it was linked to experiences of withdrawal symptoms. As this participant explains:

> *When I first started heroin my tolerance level was very low and things like that, but I find as the years go on, if I started using heroin now, smoking heroin now, I wouldn't get that feeling sick and all that and everything because for some reason my body's used–it gets used–I can get there quicker.* Interview 14

## Discussion

To the authors' knowledge this is the first study to report qualitative evidence for tolerance developing more rapidly with each relapse episode, with most participants describing that they experienced tolerance developing more rapidly compared to when they initiated drug use.

While the multi-faceted nature of tolerance has been acknowledged previously in theoretical literature [13,36], the current study addresses the paucity of evidence on the lived experience of tolerance, particularly the rate of tolerance returning following repeated episodes of abstinence [13]. In their descriptions of tolerance, participants reported both a reduced effect of the drug and a lowering of mood resulting in a reduction in the desired effect of the drug. It is conceivable therefore that tolerance to the desired effects of heroin is a combination of both pharmacological and psychological tolerance (Fig 1C) as the development of one does not preclude the other. White and Irvine [12] proposed that tolerance to opioid-induced euphoria developed to a greater extent than tolerance to respiratory depression. This means that the gap between the dose needed to get high and the dose that caused respiratory depression and overdose reduces over time [13,36]. Support for this hypothesis comes from observing that when experienced people who use heroin, who presumably had significant tolerance to its desired effects, were administered heroin respiratory depression was observed [37]. As tolerance to respiratory depression by opioids is likely to be only pharmacological in nature (Fig 1A) then, if tolerance to the desired effect of opiates involves both psychological and pharmacological

components (Fig 1C) this would be a possible explanation for differential tolerance levels between these two opioid-induced effects. Thus, using greater quantities in response to increasing tolerance to the latter, as described in this study, may increase the risk of overdose due to a mismatch in tolerance to respiratory depression. This may explain why more experienced people with greater tolerance to the drug's euphoric effects appear at greater risk of overdose than those who are less tolerant [13,36]. A disagreement may therefore exist between actual and perceived risk of overdose. Perceptions of the quality of the drug may also have a role to play. Similarly, Heavey et al. (2018) found that tolerance informed the perceived need to have naloxone available [22].

We aimed to understand experiences of tolerance to the respiratory depressant effects of heroin as well as to the desired effects, however, participants' accounts focused on the latter. Parkin et al. (2020) found 'rage' was a common negative response to naloxone administration when an overdose was occurring in part explained by participants believing it had ruined their 'high' and not realising they had been overdosing [38]. This relates to the current study's finding that participants had difficulty recognising respiratory depression. Furthermore, experimental studies on diamorphine-induced respiratory depression demonstrate extended periods of apnea (up to 56 seconds) [39] without participant awareness (personal communication, Society for the Study of Addiction Annual Conference, 2020). Participants reported that they were at higher overdose risk following abstinence, which may in part have resulted from harm reduction messages received from drug services. Analysis of respiratory depressant effects of fentanyl demonstrated that people with tolerance to opioids still experience apnea (at higher doses compared to people who are not tolerant), highlighting the overdose risk despite tolerance [4]. It is therefore important to increase awareness of differences between tolerance to the respiratory depression effects of heroin and that to its desired effects.

Participants also did not always distinguish between tolerance and dependence, highlighting the intertwined nature of these experiences. At the cellular level tolerance and physical dependence result from different neurochemical adaptive changes to opioid exposure [8,9]. That participants considered them as part of the same phenomenon may be explained by the fact that increased levels of tolerance would shorten the time to withdrawal signs appearing after taking the drug (Fig 1D). This has implications for harm reduction services in that discussions around tolerance should also consider dependence. Animal studies enable assessment of tolerance independent of dependence by maintaining the latter through constant drug administration (preventing withdrawal). Changes in opioid receptor desensitization mechanisms induced by prolonged morphine exposure have been reported in rat brain neurones [40] but whether these persist after a period of abstinence has not yet been examined. Future studies of tolerance development are needed to define the neurochemical changes that may underlie rapid tolerance development after relapse.

In this study, using heroin outside when experiencing homelessness encouraged a perpetual use-score cycle which contributes to increasing tolerance. Rushed injection practices when injecting outside is well known [41]. White and Irving have previously argued that using drugs in unfamiliar surroundings is associated with lower tolerance and higher risk of overdose [13]. We report that a small number of people try to control their tolerance levels. Similarly, Fox and colleagues found that some people use enough drugs to sustain tolerance levels during prison sentences [21]. Various forms of drug sampling are used to inform the amount of drug used, due to unknown drug strength and overdose experience [23]. For others this practice is used in response to particular situations—following phases of abstinence and when the source of heroin is new [23]. We found diverse practices among participants using heroin following a period of abstinence; while approximately half used less, others used the same amount as pre-abstinence. The latter was explained by poorer drug quality and assuming that the body could

handle the same amount. Others have found that in groups, people with higher tolerance may use drugs before those with lower tolerance and communicate the strength as a way of minimising the risk of overdose to the latter [22,23].

The potential for recall bias is a study limitation—some participants reflected on tolerance development over long periods. The similarity of experiences suggests this may not critically affect the credibility of the findings. What is presented, are reflections on perceived changes in tolerance. These insights are important because they inform behavioural responses to tolerance development and contribute toward the risk of overdose. The average age of participants (40 years) is not representative of younger people using heroin, although this is reflective of the ageing population of people who use heroin in Bristol and the UK in general [42]. Only three participants were recruited outside of the NSP, although most participants were also taking OST at the time of the interview. This means we did not speak to people who were not receiving treatment, though participants were reflecting back on experiences of being abstinent from all substances.

## Conclusions and implications

We report the first qualitative evidence of tolerance to the psychoactive effects of heroin developing more quickly with each relapse episode. This suggests some long-lasting adaptive changes have been induced by prior drug exposure and/or abstinence. Whether rapid tolerance is perceived to develop after abstinence and re-exposure to prescription opioids such as oxycodone, and to the fentanyls remains to be determined. Harm reduction messages, specifically targeted at people who use heroin, are needed to raise awareness of the different components of tolerance and the risk of overdose despite tolerance to the desired effects of the drug.

## Supporting information

**S1 Appendix. Interview topic guide.**
(DOCX)

## Acknowledgments

GH and PE conceived of the study. GH oversaw all aspects of the study including data interpretation. JK led the study design and governance procedures, data collection, analysis and write-up with input from all co-authors (EH, GH, PE, RA, MH, ST). EH led the recruitment of study participants. Data interpretation was discussed among JK, GH, MH, RA, EH. All authors contributed to and approved the content of the final manuscript.

## Author Contributions

**Conceptualization:** Pete Ellis, Graeme Henderson.

**Data curation:** Joanna May Kesten, Ed Holder.

**Formal analysis:** Joanna May Kesten, Graeme Henderson.

**Methodology:** Joanna May Kesten.

**Supervision:** Ed Holder, Rachel Ayres, Steve Taylor, Matthew Hickman, Graeme Henderson.

**Writing – original draft:** Joanna May Kesten.

**Writing – review & editing:** Ed Holder, Rachel Ayres, Pete Ellis, Steve Taylor, Matthew Hickman, Graeme Henderson.

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
