## [Decision Letter · Decision Letter 0]

2 Mar 2022

PONE-D-21-25082“The more times I have a time of abstinence and then go back to it, the quicker the tolerance level shoots right up like a thermostat on a hot day”: a qualitative study of changes in the development of opioid tolerance on re-exposurePLOS ONE

Dear Dr. Kesten,

Thank you for submitting your manuscript to PLOS ONE. After careful consideration, we feel that it has merit but does not fully meet PLOS ONE’s publication criteria as it currently stands. Therefore, we invite you to submit a revised version of the manuscript that addresses the points raised during the review process. Both reviewers found this submission of interest. However, there are major concerns expressed by reviewers regarding interpretation and approach that should be addressed. Please see attached comments provided by reviewers and provide a detailed consideration of these concerns along with your revision. Please submit your revised manuscript by Apr 16 2022 11:59PM. If you will need more time than this to complete your revisions, please reply to this message or contact the journal office at plosone@plos.org. Please include the following items when submitting your revised manuscript:A rebuttal letter that responds to each point raised by the academic editor and reviewer(s). You should upload this letter as a separate file labeled 'Response to Reviewers'.A marked-up copy of your manuscript that highlights changes made to the original version. You should upload this as a separate file labeled 'Revised Manuscript with Track Changes'.An unmarked version of your revised paper without tracked changes. You should upload this as a separate file labeled 'Manuscript'.

We look forward to receiving your revised manuscript.

Kind regards,

Juan M Dominguez, PhD

Academic Editor

PLOS ONE

Journal Requirements:

2. Please review PLOS ONE guidelines for titles at https://journals.plos.org/plosone/s/submission-guidelines#loc-parts-of-a-submission. In this case we have concern your title may not be sufficiently concise and comprehensible to readers outside of the field. Please consider these guidelines and make any changes necessary.

4. Thank you for stating in your Funding Statement: "JK and MH acknowledge support from the NIHR HPRU in Behavioural Science and Evaluation at University of Bristol and JK is partly funded by National Institute for Health Research Applied Research Collaboration West (NIHR ARC West). The views expressed are those of the author(s) and not necessarily those of the MRC, NHS England, NHS Improvement, the NIHR or the Department of Health and Social Care. ST is funded by Public Health England. GH is funded by the Medical Research Council MR/S010890/1. "

5. Thank you for stating the following in the Acknowledgments Section of your manuscript: "GH and PE conceived of the study. GH oversaw all aspects of the study including data interpretation. JK led the study design and governance procedures, data collection, analysis and write-up with input from all co-authors (EH, GH, RA, MH, ST). EH led the recruitment of study participants. Data interpretation was discussed among JK, GH, MH, RA, EH. All authors contributed to and approved the content of the final manuscript.

JK and MH acknowledge support from the NIHR HPRU in Behavioural Science and Evaluation at University of Bristol and JK is partly funded by National Institute for Health Research Applied Research Collaboration West (NIHR ARC West) at University Hospitals Bristol and Weston NHS Foundation Trust. ST is funded by Public Health England. GH is funded by the Medical Research Council MR/S010890/1. 

This study was supported by the Medical Research Council and the NIHR Health Protection Research Unit (HPRU) in Behavioural Science and Evaluation at University of Bristol, in partnership with Public Health England (PHE). The views expressed are those of the authors and not necessarily those of the MRC, NIHR, the Department of Health and Social Care, or PHE."

"JK and MH acknowledge support from the NIHR HPRU in Behavioural Science and Evaluation at University of Bristol and JK is partly funded by National Institute for Health Research Applied Research Collaboration West (NIHR ARC West). The views expressed are those of the author(s) and not necessarily those of the MRC, NHS England, NHS Improvement, the NIHR or the Department of Health and Social Care. ST is funded by Public Health England. GH is funded by the Medical Research Council MR/S010890/1. "

7. We note that you have stated that you will provide repository information for your data at acceptance. Should your manuscript be accepted for publication, we will hold it until you provide the relevant accession numbers or DOIs necessary to access your data. If you wish to make changes to your Data Availability statement, please describe these changes in your cover letter and we will update your Data Availability statement to reflect the information you provide.

Reviewers' comments:

Reviewer's Responses to Questions

**Comments to the Author**

1. Is the manuscript technically sound, and do the data support the conclusions?

Reviewer #1: No

Reviewer #2: Yes

2. Has the statistical analysis been performed appropriately and rigorously? 

Reviewer #1: N/A

Reviewer #2: N/A

3. Have the authors made all data underlying the findings in their manuscript fully available?

Reviewer #1: Yes

Reviewer #2: Yes

4. Is the manuscript presented in an intelligible fashion and written in standard English?

Reviewer #1: Yes

Reviewer #2: Yes

5. Review Comments to the Author

Reviewer #1: This is an interesting report on dependent heroin users subjective experience of tolerance. For me, the important reported themes were that for many respondents, tolerance and dependence were essentially the same; and that tolerance, which initially took time to develop, increased very rapidly after periods of abstinence. These experiences underly that tolerance is central to dependence, and are consistent with neuroscientific evidence that dependence produces lasting brain changes rendering people vulnerable to relapse. They are not new observations .The study also noted that users were unaware of (changes in) respiratory response to opioids.

However, I have some problems with the way the report is framed.

The conclusion “By elucidating the mechanism(s) involved and potentially discovering how they could be switched on prior to relapse occurring we might be able to develop a beneficial harm reduction treatment for people in abstinence that would protect against overdose occurring on subsequent relapse” is not something I can derive from the themes. Surely, educating drug users about loss of tolerance during abstinence? Or that the paper provides experiential support for the finding of lasting brain changes following dependence? These modest conclusions might be more appropriate.

I also have difficulty with the statement that “at a rudimentary level we can divide tolerance mechanisms in two: biological and psychological (6)”. Reference 6 identifies pharmacokinetic, pharmacodynamic and suggests there is “learned” tolerance. If this is what the authors are referring to as “psychological” tolerance, they should use the term “learned”. There are complex biological substrates of mental events, and distinguishing biological and psychological seems to me meaningless, but the distinction runs through the introduction and discussion. The discussion includes the claim “As tolerance to respiratory depression by opioids is likely to be only pharmacological in nature then, if tolerance to the desired effect of opiates involves both psychological and pharmacological components this would be a possible explanation for differential tolerance levels between these two opioid-induced effects.” This speculation is not related to the qualitative data. Creating a dubious distinction between psychological and biological factors seems irrelevant in a qualitative study of users’ subjective experiences.

Introduction, 3rd paragraph, first line “psychotomimetic” means simulating psychosis, and is the incorrect word; simplest to say “subjective”.

Authors say that “White and Irvine (11) suggested in their theoretical analysis that tolerance to opioid induced ‘euphoria’ developed to a greater extent than tolerance to respiratory depression although there is a lack of experimental evidence to substantiate this theory.” But this phenomenon has been observed experimentally, as noted in the discussion (ref 34).

On p13 the authors cite on respondent’s comments as evidence of a psychological component to tolerance; but what the respondent is actually reporting is fear of withdrawal – and learned (conditioned) withdrawal was first described in 1948 by Wikler.

Reviewer #2: Dear Authors

Thank you for giving me the opportunity to read this fascinating paper. A strength of qualitative research is that it can lead you to think about thing in a completely different way and this one is no exception - in particular the insights that

1.) some patients view tolerance as influence on duration of action and relate it to withdrawal rather than peak effect

2.) patients do not perceive respiratory depression, or changes in this

This paper will change the way I talk to my patients about tolerance, and so from my perspective it is of considerable interest and clinical relevance. I also note that the research team is a translational group and part of the stated aim is to inform preclinical studies of tolerance, which neatly addresses some of the problems with lack of correspondence between animal models and the clinical situation.

It is clearly and engagingly written.

Suggestions for improvement:

1.) Qualitative research tends to be underpinned by particular epistemology or theoretical frameworks and this informs how we interpret the data. I couldn't identify much about where the authors were coming from in this regard. Please could the authors clarify their framework and how this influenced design and interpretation.

2.) The authors state that participants were recruited from BDP a large urban provider with multiple sites, but they do not specify what proportion of participants came from where. It would indicate a relative strength of the study if participants were drawn from different locations rather than a single needle exchange, for example, and this is not reported.

3.) Table 1 could be presented in a more conventional way ie with median and range for age, age of first use of heroin, and duration of abstinence. Median abstinence duration is particularly important given the massive range.

5.) Themes: I think the authors could emphasise more the surprising finding that some users think about tolerance in a completely different way to clinicians! The longitudinal conception of tolerance and thinking about it as how long the effect lasts before you get withdrawal rather than exclusively around intensity of high (or from a clinician's perspective, severity of respiratory depression) was really interesting. For me this was less surprising/novel then the finding that tolerance develops more quickly following a period of abstinence.

Minor stuff:

Page 4: Please use either ‘detoxification’ or ‘medically-assisted withdrawal’ instead of detox.

Page 6 last para – in clinical practice it is commonly more than three days missed, not five. I note the nuance in the UK guidelines but in practice the pharmacy will refuse to give the fourth dose if the patient hasn’t attended- suggest missing three doses prompts re-assessment and – really – five days would warrant re-titration from a starting dose.

6. PLOS authors have the option to publish the peer review history of their article (what does this mean?). If published, this will include your full peer review and any attached files.

Reviewer #1: **Yes: **James Bell

Reviewer #2: No

---

## [Author Response · Author response to Decision Letter 0]

5 May 2022

Dear Juan M Dominguez and referees, 

We are grateful for the opportunity to resubmit this paper and appreciate the detailed feedback given. 

We have responded to each point below and highlighted changes in the manuscript in red. We believe the manuscript has been strengthened as a result of responding to the feedback. 

Kind regards,

Jo Kesten (on behalf of all co-authors)

Apologies, we have amended the manuscript to meet PLOS ONE’s style requirements. 

2. Please review PLOS ONE guidelines for titles at https://journals.plos.org/plosone/s/submission-guidelines#loc-parts-of-a-submission. In this case we have concern your title may not be sufficiently concise and comprehensible to readers outside of the field. Please consider these guidelines and make any changes necessary.

Thank you, we have made the title more concise and understandable by removing the quote and adding the population group: “Changes in the development of opioid tolerance on re-exposure among people who use heroin: a qualitative study”

4. Thank you for stating in your Funding Statement: "JK and MH acknowledge support from the NIHR HPRU in Behavioural Science and Evaluation at University of Bristol and JK is partly funded by National Institute for Health Research Applied Research Collaboration West (NIHR ARC West). The views expressed are those of the author(s) and not necessarily those of the MRC, NHS England, NHS Improvement, the NIHR or the Department of Health and Social Care. ST is funded by Public Health England. GH is funded by the Medical Research Council MR/S010890/1. "

5. Thank you for stating the following in the Acknowledgments Section of your manuscript: "GH and PE conceived of the study. GH oversaw all aspects of the study including data interpretation. JK led the study design and governance procedures, data collection, analysis and write-up with input from all co-authors (EH, GH, RA, MH, ST). EH led the recruitment of study participants. Data interpretation was discussed among JK, GH, MH, RA, EH. All authors contributed to and approved the content of the final manuscript.

JK and MH acknowledge support from the NIHR HPRU in Behavioural Science and Evaluation at University of Bristol and JK is partly funded by National Institute for Health Research Applied Research Collaboration West (NIHR ARC West) at University Hospitals Bristol and Weston NHS Foundation Trust. ST is funded by Public Health England. GH is funded by the Medical Research Council MR/S010890/1. 

This study was supported by the Medical Research Council and the NIHR Health Protection Research Unit (HPRU) in Behavioural Science and Evaluation at University of Bristol, in partnership with Public Health England (PHE). The views expressed are those of the authors and not necessarily those of the MRC, NIHR, the Department of Health and Social Care, or PHE."

"JK and MH acknowledge support from the NIHR HPRU in Behavioural Science and Evaluation at University of Bristol and JK is partly funded by National Institute for Health Research Applied Research Collaboration West (NIHR ARC West). The views expressed are those of the author(s) and not necessarily those of the MRC, NHS England, NHS Improvement, the NIHR or the Department of Health and Social Care. ST is funded by Public Health England. GH is funded by the Medical Research Council MR/S010890/1. "

We have removed the funding information from the acknowledgements section. 

The funding section should read as follows: 

JK and MH acknowledge support from the NIHR HPRU in Behavioural Science and Evaluation at University of Bristol and JK is partly funded by National Institute for Health Research Applied Research Collaboration West (NIHR ARC West) at University Hospitals Bristol and Weston NHS Foundation Trust. ST is funded by Public Health England. GH is funded by the Medical Research Council MR/S010890/1. 

This study was funded by the Medical Research Council and the NIHR Health Protection Research Unit in Behavioural Science and Evaluation at University of Bristol, in partnership with UK Health Security Agency (UKHSA). The views expressed are those of the author and not necessarily those of the MRC, the NIHR, the Department of Health and Social Care, or UKHSA. There was no additional external funding received for this study.

Thank you the data are now available at the University of Bristol data repository, data.bris, at https://doi.org/10.5523/bris.23rgdwwmdtix52gqa51ith8gxy. 

7. We note that you have stated that you will provide repository information for your data at acceptance. Should your manuscript be accepted for publication, we will hold it until you provide the relevant accession numbers or DOIs necessary to access your data. If you wish to make changes to your Data Availability statement, please describe these changes in your cover letter and we will update your Data Availability statement to reflect the information you provide.

Please see above. 

Reviewer 1 comments Response notes

Reviewer #1: This is an interesting report on dependent heroin users subjective experience of tolerance. For me, the important reported themes were that for many respondents, tolerance and dependence were essentially the same; and that tolerance, which initially took time to develop, increased very rapidly after periods of abstinence. These experiences underly that tolerance is central to dependence, and are consistent with neuroscientific evidence that dependence produces lasting brain changes rendering people vulnerable to relapse. They are not new observations .The study also noted that users were unaware of (changes in) respiratory response to opioids.

 Thank you for your positive response to our manuscript. 

We thank the referee for highlighting the important relationship between tolerance and physical dependence. This relationship is now discussed in more detail in the Discussion (lines 456 to 460) and a new schematic Fig. 1D.

Regarding the development of tolerance we are unaware of any previously published research studies relating to the rate of tolerance development being faster after periods of abstinence and therefore believe this to be a new observation and hypothesis raised by the qualitative study that may encourage others to consider this phenomenon in their future research on the importance of tolerance development to different effects of opioids.

The conclusion “By elucidating the mechanism(s) involved and potentially discovering how they could be switched on prior to relapse occurring we might be able to develop a beneficial harm reduction treatment for people in abstinence that would protect against overdose occurring on subsequent relapse” is not something I can derive from the themes. Surely, educating drug users about loss of tolerance during abstinence? Or that the paper provides experiential support for the finding of lasting brain changes following dependence? These modest conclusions might be more appropriate. We agree fully with the referee that the loss of tolerance during abstinence is an important harm reduction message. It is already a major emphasis in harm reduction.

Our discussion of the potential for harm reduction concerning faster development of tolerance after abstinence relates to the pre-clinical nature of our ongoing research. See page 6, lines 103-106: “We wanted this work to inform / run alongside a lab based study: our future goal is to undertake reverse translational research (using animal experiments) to define these neurochemical changes in the brain, it is pertinent to first understand the lived experience of tolerance development over time and following relapse (22) and then build upon that to investigate potential biological mechanisms.”

The importance of this aspect was highlighted by Reviewer 2 who stated ‘I also note that the research team is a translational group and part of the stated aim is to inform preclinical studies of tolerance…’

The referee’s recommended conclusions can be found on p24, lines 497-499 specifying the need for harm reduction messages: “Harm reduction messages, specifically targeted at people who use heroin, are needed to raise awareness of the different components of tolerance and the risk of overdose despite tolerance to the desired effects of the drug.”

I also have difficulty with the statement that “at a rudimentary level we can divide tolerance mechanisms in two: biological and psychological (6)”. Reference 6 identifies pharmacokinetic, pharmacodynamic and suggests there is “learned” tolerance. If this is what the authors are referring to as “psychological” tolerance, they should use the term “learned”. There are complex biological substrates of mental events, and distinguishing biological and psychological seems to me meaningless, but the distinction runs through the introduction and discussion. The discussion includes the claim “As tolerance to respiratory depression by opioids is likely to be only pharmacological in nature then, if tolerance to the desired effect of opiates involves both psychological and pharmacological components this would be a possible explanation for differential tolerance levels between these two opioid-induced effects.” This speculation is not related to the qualitative data. Creating a dubious distinction between psychological and biological factors seems irrelevant in a qualitative study of users’ subjective experiences. Please see previous point about the intention to inform preclinical studies of tolerance.

In response to the reviewer, we have removed the term ‘biological’ and replaced it with ‘pharmacological’ to encompass pharmacokinetic and pharmacodynamic processes.

We believe that there are a number of adaptive mechanisms that contribute to the ‘psychological’ component of opioid tolerance. As the referee correctly points out, one of these is ‘learned’ but other adaptive responses that give rise to negative emotional states are also involved. We now define in greater detail in the Introduction what we mean by a ‘psychological’ component. 

White and Irvine have suggested that tolerance increases to the euphoric effect of opioids more quickly than to their respiratory depressant effect. If after abstinence tolerance to euphoria develops faster than previously but tolerance to respiratory depression does not then that would increase even further the risk of overdose. To determine if this is the case requires further studies and in our discussion we have sought to highlight that need. We also thought it relevant to propose a potential explanation for why tolerance might develop less for respiratory depression in Fig 1. 

Introduction, 3rd paragraph, first line “psychotomimetic” means simulating psychosis, and is the incorrect word; simplest to say “subjective”. We apologise for this error. We had meant to use ‘psychoactive’ as we have done elsewhere in the manuscript. This has now been corrected.

Authors say that “White and Irvine (11) suggested in their theoretical analysis that tolerance to opioid induced ‘euphoria’ developed to a greater extent than tolerance to respiratory depression although there is a lack of experimental evidence to substantiate this theory.” But this phenomenon has been observed experimentally, as noted in the discussion (ref 34). We apologise for any confusion, the paper referred to (ref 34) did not study changes in the degree of tolerance to respiratory depression compared to ‘euphoria’ but only commented on the fact they could observe respiratory depression in heroin users and that is consistent with the hypothesis of White and Irvine that tolerance to respiratory depression might be less. We have removed the word ‘Experimental’ on line 427 to clarify this. We believe that there is a need for a thorough study of the rate of development and degree of tolerance to respiratory depression compared to ‘euphoria’ but there may be ethical issues involved in doing such a study in humans. 

On p13 the authors cite one respondent’s comments as evidence of a psychological component to tolerance; but what the respondent is actually reporting is fear of withdrawal – and learned (conditioned) withdrawal was first described in 1948 by Wikler We thank the referee for highlighting this error. We agree that the participant is referring to fear of withdrawal. We have replaced the quote (lines 228 - 232).

Reviewer #2 comments Response notes 

Thank you for giving me the opportunity to read this fascinating paper. A strength of qualitative research is that it can lead you to think about things in a completely different way and this one is no exception - in particular the insights that

1.) some patients view tolerance as influence on duration of action and relate it to withdrawal rather than peak effect

2.) patients do not perceive respiratory depression, or changes in this

This paper will change the way I talk to my patients about tolerance, and so from my perspective it is of considerable interest and clinical relevance. I also note that the research team is a translational group and part of the stated aim is to inform preclinical studies of tolerance, which neatly addresses some of the problems with lack of correspondence between animal models and the clinical situation.

It is clearly and engagingly written.

 Thank you for your encouraging response to our manuscript. We are pleased to hear that this research is valuable to clinical practice and will make a difference. 

In response to the importance of the finding that participants view tolerance and dependence as synonymous we have added this to the abstract and provide more detail in the Discussion along with a new schematic in Figure 1D. 

Yes, as the referee correctly notes, the rationale for this paper is to drive preclinical studies where both tolerance and withdrawal can be examined simultaneously.

1.) Qualitative research tends to be underpinned by particular epistemology or theoretical frameworks and this informs how we interpret the data. I couldn't identify much about where the authors were coming from in this regard. Please could the authors clarify their framework and how this influenced design and interpretation. Thank you for this feedback. We have included the following in the methods to clarify our epistemological position on lines 151-153: 

“The study adopted an interpretative approach seeking to understand how people interpret and perceive their experiences and the social world and explore the actions people take in response to these experiences (29).”

2.) The authors state that participants were recruited from BDP a large urban provider with multiple sites, but they do not specify what proportion of participants came from where. It would indicate a relative strength of the study if participants were drawn from different locations rather than a single needle exchange, for example, and this is not reported. Thank you. Our recruitment approach aimed to achieve a diverse sample in relation to experiences of abstinence, relapse and tolerance. We did not sample to achieve a wide geographical spread and did not record this information. Recruitment took place through a needle and syringe programme and an OST service run from GP practices across the city indicating there is likely to be geographical diversity though we do not view this as important in shaping the experiences under study. 

3.) Table 1 could be presented in a more conventional way ie with median and range for age, age of first use of heroin, and duration of abstinence. Median abstinence duration is particularly important given the massive range. We have updated age, age using heroin and number of abstinence episodes to the suggested format.

Unfortunately, length of abstinence cannot be presented as an average because some participants did not give an exact timeframe.

5.) Themes: I think the authors could emphasise more the surprising finding that some users think about tolerance in a completely different way to clinicians! The longitudinal conception of tolerance and thinking about it as how long the effect lasts before you get withdrawal rather than exclusively around intensity of high (or from a clinician's perspective, severity of respiratory depression) was really interesting. For me this was less surprising/novel then the finding that tolerance develops more quickly following a period of abstinence. Thank you, we’ve enhanced the discussion in relation to this including a part to Fig 1. 

Page 4: Please use either ‘detoxification’ or ‘medically-assisted withdrawal’ instead of detox. Thank you, we have replaced ‘detox’ with ‘detoxification’. We do not wish to use the term ‘medically assisted withdrawal’ because on some occasions some of the participants stopped using opioids without medical assistance. 

Page 6 last para – in clinical practice it is commonly more than three days missed, not five. I note the nuance in the UK guidelines but in practice the pharmacy will refuse to give the fourth dose if the patient hasn’t attended- suggest missing three doses prompts re-assessment and – really – five days would warrant re-titration from a starting dose. We agree that shorter timeframes are often used clinically. We adhered to the clinical guidelines (which are longer) to make sure that participants had been abstinent for long enough for tolerance to have declined significantly.

---

## [Editor Report · Decision Letter 1]

20 May 2022

Changes in the development of opioid tolerance on re-exposure among people who use heroin: a qualitative study

PONE-D-21-25082R1

Dear Dr. Kesten,

We’re pleased to inform you that your manuscript has been judged scientifically suitable for publication and will be formally accepted for publication once it meets all outstanding technical requirements.

Kind regards,

Juan M Dominguez, PhD

Academic Editor

PLOS ONE
---

## [Editor Report · Acceptance letter]

14 Jun 2022

PONE-D-21-25082R1 

Changes in the development of opioid tolerance on re-exposure among people who use heroin: a qualitative study 

Dear Dr. Kesten:

I'm pleased to inform you that your manuscript has been deemed suitable for publication in PLOS ONE. Congratulations! Your manuscript is now with our production department. 

Kind regards, 

on behalf of

Dr Juan M Dominguez 

Academic Editor

PLOS ONE